# Optimization, Nature, and Mechanism Investigations for the Adsorption of Ciprofloxacin and Malachite Green onto Carbon Nanoparticles Derived from Low-Cost Precursor via a Green Route

**DOI:** 10.3390/molecules27144577

**Published:** 2022-07-18

**Authors:** Rasmiah S. Almufarij, Babiker Y. Abdulkhair, Mutaz Salih, Haia Aldosari, Najla W. Aldayel

**Affiliations:** 1Department of Chemistry, College of Science, Princess Nourah Bint Abdulrahman University, P.O. Box 84428, Riyadh 11671, Saudi Arabia; rsmufrrig@pnu.edu.sa (R.S.A.); nwaldayel@gmail.com (N.W.A.); 2Department of Chemistry, College of Science, Imam Mohammad Ibn Saud Islamic University (IMSIU), P.O. Box 90905, Riyadh 11623, Saudi Arabia; 3Department of Chemistry-Hurrymilla, College of Science and Humanities, Imam Mohammad Ibn Saud Islamic University (IMSIU), P.O. Box 5701, Hurrymilla 11432, Saudi Arabia; meamin@imamu.edu.sa; 4Department of Physics, College of Science, Shaqra University, P.O. Box 5701, Shaqra 11961, Saudi Arabia; haldossari@su.edu.sa

**Keywords:** carbon nanoparticles, sunflower seed waste, ball milling, ciprofloxacin, malachite green

## Abstract

The spread of organic pollutants in water spoils the environment, and among the best-known sorbents for removing organic compounds are carbonaceous materials. Sunflower seed waste (SFSW) was employed as a green and low-cost precursor to prepare carbon nanoparticles (CNPs) via pyrolysis, followed by a ball-milling process. The CNPs were treated with a nitric–sulfuric acid mixture (1:1) at 100 °C. The scanning electron microscopy (SEM) showed a particle size range of 38 to 45 nm, and the Brunauer–Emmett–Teller (BET) surface area was 162.9 m^2^ g^−1^. The elemental analysis was performed using energy-dispersive X-ray spectroscopy, and the functional groups on the CNPs were examined with Fourier transform infrared spectroscopy. Additionally, an X-ray diffractometer was employed to test the phase crystallinity of the prepared CNPs. The fabricated CNPs were used to adsorb ciprofloxacin (CFXN) and malachite green (MLG) from water. The experimentally obtained adsorption capacities for CFXN and MLG were 103.6 and 182.4 mg g^−1^, respectively. The kinetic investigation implied that the adsorption of both pollutants fitted the pseudo-first-order model, and the intraparticle diffusion step controlled the process. The equilibrium findings for CFXN and MLG sorption on the CNPs followed the Langmuir and the Fredulich isotherm models, respectively. It was concluded that both pollutants spontaneously adsorbed on the CNPs, with physisorption being the likely mechanism. Additionally, the FTIR analysis of the adsorbed CFXN showed the disappearance of some functional groups, suggesting a chemisorption contribution. The CNPs showed an excellent performance in removing CFXN and MLG from groundwater and seawater samples and possessed consistent efficiency during the recycle–reuse study. The application of CNPs to treat synthetically contaminated natural water samples indicated the complete remediation of polluted water using the ball-mill-fabricated CNPs.

## 1. Introduction

Water pollution is a major environmental problem facing the world today. The World Health Organization (WHO) and the United Nations International Children’s Emergency Fund (UNICEF) reported the non-accessibility of ten million persons to adequate water sources. The decay of the water environment may be caused by the accidental or illegal discharge of polluted effluents [1,2,3,4]. Malachite green (MLG; Figure 1a) is among some of the most dangerous organic pollutants that damage aquatic environments [5]. Although it is toxic and carcinogenic, MLG is still used in various industries, including food and pharmaceuticals [5,6,7,8,9,10]. A recent study revealed that 70% of antibiotic doses were expelled in urine and feces into wastewater [6]. Pharmaceutical pollutants (PhPs) have been detected in sewage, water resources, and even in the tap water of some countries [7,8,9,10]. In addition to the typical risk of PhPs in water, antibiotics might enhance microbe immunity, causing mutations and, hence, generating new diseases [11]. Ciprofloxacin (CFXN; Figure 1b) is the most prescribed drug among its antibiotic family [12,13,14].

Numerous treatment methodologies have failed to clear water from PhPs and dyes, including bioremediation and ozonation [15,16]. On the other hand, photocatalysis and adsorption processes appear to have promising results in removing such pollutants [17,18,19,20]. Researchers are still in disagreement about the superiority of adsorption over photodegradation, but it is undisputed that the adsorption process consumes less energy. Because of their hazardous occurrence in water, several sorbents have been tested for removing MLG and CFXN from polluted water [21,22,23,24,25]. Historically, carbonaceous materials (CMs) are known for their high surface areas, excellent adsorption capacity, and robustness. Additionally, their surface can be modified from nonpolar to polar to suit the characteristic diversity of compounds [26,27,28,29,30,31,32,33,34]. CMs can be fabricated from renewable green agricultural wastes, which can be superabundant, nontoxic, and inexpensive. The circular economy and green chemistry concepts emphasize the exploitation–recycling of agricultural waste into valuable products. The United States Department of Agriculture (USDA) stated that approximately 46 million tons of sunflower seeds are produced globally [35]. The edible oil industries generate sunflower seed waste (SFSW), and it would be helpful to benefit from such a massive waste.

This study aims to eliminate SFSW by employing it as a cheap green precursor for producing carbon nanoparticles (CNPs) via a simple route. Since the adsorption efficiency is affected by sorbent and sorbate natures, the prepared CNPs are functionalized to suit removing polar PhPs and organic dyes, exemplified by CFXN and MLG, respectively.

## 2. Experimental Section

### 2.1. Materials

Nitric acid, 72% (HNO_3_), and sulfuric acid, 98% (H_2_SO_4_), were supplied by Sigma-Aldrich, St. Louis, MO, USA. The SFSW was collected from the local market (KSA). The MLG was provided by LO–BACHEM, India, and the CFXN was provided by Rhanboxy, Mumbai, India. Seawater and groundwater samples (SWSs and GWSs) were employed to test the CNPs’ applicability in removing such pollutants from natural contaminated water resources.

### 2.2. Fabrication of CNPs

Approximately 50.0 g of SFSW was transferred to a boat crucible, placed into a tubular furnace, and carbonized at 600 °C under a nitrogen stream for two hours. The CMs were placed in a stainless-steel crucible of a vertical planetary ball mill (model: Al-VPB; Mumbai, India) and operated at 500 RPM for 10.0 h. An amount of 10.0 g of the obtained powder was stirred with 100 mL of HNO_3_: H_2_SO_4_ mixture (1:1) for one hour at 100 °C. The product was dispersed in 200 mL of distilled water and sonicated for 30 min. The resultant nanoparticles were filtered off, washed, and dried at 110 °C for three hours.

### 2.3. Characterization of the CNPs

The surficial and detailed morphologies of CNPs were investigated using scanning electron–energy-dispersive microscopy (SEM-EDX, JSM-IT300) and a transmission electron microscope (TEM, JEM-1400). The detailed morphology was examined via transmission electron microscopy. In addition, the functional groups on the surface of CNPs were analyzed with Fourier transform infrared spectroscopy (FTIR; Bruker TENSOR-FTIR). Additionally, the surface characteristics were analyzed via an ASAP 2020 Micromeritics surface. The sample was placed in a quartz tube, vacuumed to 10 μm Hg at 150 °C for 6.0 h, and then analyzed using ultrapure nitrogen gas (N_2_).

The nanoparticles’ phase purity was analyzed using a powder X-ray diffractometer (XRD) (Bruker, D8 Advance; Billerica, MA, USA). The average crystal size of the CNPs was calculated via Scherrer’s equation (Equation (2)), and the lattice parameters (*a* and *c*) and the lattice imperfection (*ε*) were estimated via Equations (3)–(5), respectively [36,37].
(1)$$D=\frac{0.9\lambda }{\beta \cos\theta }$$
(2)$$a=\frac{\lambda }{\sqrt{3}\;sin\theta }$$
(3)$$c=\frac{\lambda }{sin\theta }$$
(4)$$\varepsilon =\frac{\beta }{4\;cos\theta }\;$$
where *θ*, *λ*, and *β* represent the Bragg’s angle, *Cu Kα* line (1.5406 Ǻ), and the peak width at half-maximum, respectively, while *D* represents the computed crystal size [38].

### 2.4. Kinetics and Solution Parameters

A batch procedure was employed in testing the adsorption of MLG and CFXN on the fabricated CNPs from water. Typically, 50 mg of CNPs was mixed with 120 mL of 50 mg L^−1^ pollutant aqueous solution. In addition, the pH influence was inspected by adjusting the pollutant solution to a different pH in the range of 2 to 11. The sorption process was followed by monitoring the absorbance of MLG and CFXN using a UV–Vis spectrophotometer (Shimadzu-2600i, Japan) at 617 and 273 nm, respectively. The adsorption capacity for each pollutant on the CNPs (*q_t_*, mg g^−1^) was calculated via Equation (1).
(5)$$q_{t}=\frac{\left(C_{o}-C_{t}\right)\;v}{m}$$

*C_o_* and *C_t_* (mg L^−1^) are the pollutant concentrations at time zero and *t* (min); *v* and m are the solution volume (mL) and adsorbent mass (g), respectively.

### 2.5. Isotherms and Thermodynamics

The impact of concentration and temperature was investigated by conducting sorption processes within concentration ranges of 10 to 50 mg L^−1^ CFXN and 10 to 100 mg L^−1^ MLG solutions at 20 °C, 35 °C, and 50 °C, respectively. The obtained results were utilized for the isotherm and thermodynamic investigations.

## 3. Results and Discussion

### 3.1. Characterization of the Fabricated CNPs

The SEM analysis was utilized to investigate the surface morphology of the as-prepared CNPs. A 25,000 magnification showed large clumps of approximately 200 nm and tiny particles (Figure 2a), while a 100,000 times magnification revealed that the lumps were clustered nanoparticles with a size of 38 to 45 nm (Figure 2b). These findings indicated a successful fabrication of CNPs from SFSW via calcination, followed by a ball-milling process. Moreover, the EDX analysis was employed to examine the elemental composition of the fabricated CNPs (Figure 2c,d). The analysis revealed that the CNPs were composed mainly of carbon and oxygen, and that the acid treatment had implanted oxygen groups all over the CNPs. The TEM results for the SFSW CNPs revealed that the 600 °C produced a crystal form of carbon (Figure 2e). Additionally, Figure 2f shows a group of carbon nanoparticles of approximately 10 to 20 nm in addition to minor large particles, which was in line with the SEM findings.

Figure 3a illustrates the XRD pattern for the fabricated CNPs. The diffraction peaks at 2θ◦ of 26.02 and 43.42 could be assigned to (002) and (100) of the cubical lattice graphite phase (JCPDS no. 04-0850) [39,40]. In addition, the absence of the amorphous carbon diffraction peak at 2θ° of 11.7 indicated a good crystallinity for the prepared CNPs. The D, a, and c values for the CNP crystals were 25.05, 0.26, and 5.52 nm, respectively, while the crystal imperfection value was 0.37 (a.u).

Furthermore, the surficial functional groups of the prepared CNPs were surveyed via the FTIR analysis (Figure 3b). The peaks at 850 and 2850 cm^−1^ could be assigned to a long chain’s out-of-plane C-H wagging and C-H stretching vibrations. The peaks at 1120, 1760, and 3270 cm^−1^ could be attributed to C-O, C=O, and an intermolecular hydrogen-bonded O-H stretching vibration. The gathering of these findings inferred a successful implanting of carboxylic groups on CNP surfaces. The band at 1650, 2010, and 2280 cm^−1^ could be assigned to C=C, carbon skeleton accumulated C=C, and C≡C stretching vibrations. Additionally, the 3270 and 3650 cm^−1^ bands could be set to an acid O-H stretching vibration with intramolecular H-bonding and the alcoholic O-H stretching vibration without H-bonding, respectively [36,41].

The N_2_ adsorption–desorption technique was utilized to investigate the surface features of the as-synthesized CNPs. The surface area (SA) was determined via the Brunauer–Emmett–Teller (BET) method, and the obtained isotherm and pore distribution are shown in Figure 3c,d. The CNPs demonstrated a type (III) hysteresis loop belonging to mesoporous materials [42]. The Barrett–Joyner–Halenda (BJH) method was utilized to estimate the pore diameter, width, and volume (PD, PW, and PV). The BET SA was 162.92 m^2^ g^−1^, while the PD, PW, and PV were 77.82 Ǻ, 23.11 Ǻ, and 0.10 cm^3^ g^−1^, respectively.

### 3.2. Adsorption Investigation

Figure 4a presents the contact time study for the adsorption of CFXN and MLG on the CNPs. The adsorption of CFXN from water by the CNPs reached its equilibrium point in 60 min, while MLG required 120 min. Additionally, the initial fed concentration was a crucial factor affecting adsorption. Figure 4b,c demonstrate a direct proportionality between the fed concentration and the obtained *q_t_*, which reached 103.6 and 182.4 mg g^−1^ from the 50 mg L^−1^ CFXN and 100 mg L^−1^ MLG solutions. Increasing the initial concentration could have generated an efficient force that facilitated the migration of pollutants. On the contrary, the raising of the solution’s temperature was inversely proportional to the *q_t_* of CFXN and MLG, indicating exothermic sorption (Figure 4b,c). Furthermore, the *q_t_* proportionately with fed concentrations inferred the suitability of a 5:12 ratio of sorbent mass to solution volume within the tested concentration ranges. These findings of fast uptake, short equilibrium time, and relatively high experimental *q_t_* values were competitive with recent CMs in the literature [17,43,44,45,46].

Considering the chemical structure of MLG and CFXN (Figure 1), electrostatic attraction plays a significant role in removing these pollutants [47,48]. At low pH, the auxochrome group of MLG (pKa = 10.3) is protonation, resulting in a positive charge density [49]. Additionally, the CFXN (pKa = 5.9) molecule is no less complex, as it has both acidic and amino groups. Adding to that, the implanted oxygen groups of the functionalized CNPs may explain the influence of pH on the availability and accessibility of the functional groups on the sorbent’s surface and pollutants. Figure 4d illustrates the impact of altering the solution’s pH on the *q_t_* values for CFXN and MLG removal with the prepared CNPs. Both pollutants were better removed at a pH value of 6.0, and their *q_t_* values decreased significantly below pH 5.0 and above pH 8.0. The highly available H^+^ may protonate the electron-rich sites on the pollutants and/or CNPs at low pHs. On the other hand, the ^−^OH availability above a pH of 8.0 may compete with contaminants on the adsorption sites of the sorbent [50].

### 3.3. Adsorption Kinetics

The adsorption kinetics of CFXN and MLG adsorption on the ball-mill-fabricated CNPs were investigated. The adsorption rate order was examined via the pseudo-first-order models (PSFO) and the pseudo-second-order model (PSSO) expressed in Equations (6) and (7). Additionally, an examination of the step controlling the adsorption was conducted by employing the liquid-film diffusion model (LFDM) (Equation (8)) and the intraparticle diffusion model (IPDM) (Equation (9)) [51].
(6)$$ln\left(q_{e}-q_{t}\right)=ln\;q_{e}-k_{1}\cdot t\;$$
(7)$$\frac{1}{q_{t}}=\frac{1}{k_{2}\cdot q_{e\;}^{2\;}t}+\frac{1}{q_{e}}\;$$
(8)$$q_{t}=K_{IP}*t^{\frac{1}{2}}+C_{i}\;$$
(9)$$ln\left(1-F\right)=-K_{LF}*t\;$$
where *q_e_* (mg g^−1^) represents *q_t_* at equilibrium; *k*_1_ (min^–1^) and *k*_2_ (g mg^–1^ min^–1^) are the rate adsorption constants for the PSFO and PSSO models, which were calculated from the slope and intercept values, respectively. The LFDM and IPDM constants are represented as *K_IP_* (mg g^−1^ min^−0^.^5^) and *K_LF_* (min^–1^), respectively, and both were computed from their slope values. *C_i_* (mg g^−1^) is a boundary layer thickness factor [52,53].

Figure 5a illustrates the PSFO regression lines for CFXN and MLG adsorption on the CNPs, while Figure 5b shows their PSSO linear plots. The R^2^ values of the PSFO were 0.945 and 0.970 for CFXN and MLG, while the PSSO possessed R^2^ values of 0.862 and 0.857, respectively. These findings revealed that CFXN and MLG sorption on the ball-mill CNPs followed the PSFO model [54].

Figure 5c,d present the LFDM and IPDM investigations for CFXN and MLG removal using the CNPs. The CFXN sorption exhibited K_LF_ and K_IP_ values of 0.050 min^–1^ and 2.743 mg g^−1^ min^−1/2^, respectively, and R^2^ values of 0.866 and 0.960, respectively. Additionally, the adsorption mechanism investigation for the MLG showed K_LF_ and K_IP_ values of 0.043 min^–1^ and 2.720 mg g^−1^ min^−1/2^, respectively, with R^2^ values of 0.774 and 0.983, respectively. These results indicated that the adsorption of both pollutants on the CNPs fitted the IPDM with a good agreement.

### 3.4. Adsorption Isotherms

The possibility of a monolayer sorption for CFXN and MLG using the CNPs was inspected with the Langmuir isotherm (LI) model (Equation (10)). Additionally, the multilayer adsorption of CFXN and MLG using CNPs was examined via the Freundlich isotherm (FI) model (Equation (11)) [55,56,57].
(10)$$\frac{1}{q_{e}}=\frac{1}{K_{L}{\;q}_{m}}.\;\frac{1}{C_{e}}+\frac{1}{K_{L}}$$
(11)$$\ln q_{e}=\ln K_{F}+\frac{1}{n}\;\ln C_{e}$$
where K_L_ (L mg^−1^) and K_F_ (L mg^−1^) are the LI and FI constants; q_m_ (mg g^−1^) is the possibly maximum q_t_; C_e_ (mg L^−1^) is the equilibrium pollutant’s concentration; n is the Freundlich heterogeneity factor.

For the LI model (Figure 6a,b), the reciprocal of the slope resulted in the K_L_ value, which was applied with the intercept value to compute q_m_ (Table 1). Additionally, the K_F_ value of the FI model (Figure 6c,d) was equal to the anti-LN of the obtained intercept, while the 1/n was equal to the resulting slope value (Table 1).

The adsorption of CFXN on the CNPs showed a better fitting to the LI model, while the MLG adsorption followed the FI model. Although the 1/n values indicated favorable adsorption for both pollutants, the 1/n value for CFXN was double that of the MLG, indicating that CFXN sorption had less preference for the FI model [57]. Additionally, the 1/n values for CFXN and MLG on CNPs indicated reversible adsorption, which may imply a physisorption nature.

### 3.5. Thermodynamics

The thermodynamics of CFXN and MLG adsorptions on the fabricated CNPs were investigated.

The adsorbed and remaining concentrations (*C*_ad,_ and *C*_e_, (mg L^−1^)) were employed for computing the equilibrium constant *K*_c_. The enthalpy (∆*H**^o^*, (kJ mol^−^^1^)) value was calculated using the slope from the plot of Equation (12), and the entropy (∆*S**^o^*, (kJ mol^−^^1^)) was calculated using the intercept value (Figure 7). The Gibbs free energy (∆*G**^o^*, (kJ mol^−^^1^)) was computed by applying the results from Equation (12) in Equation (13), yielding the results shown in Table 2.
(12)$$ln\;K_{c}=\;\frac{\Delta H^{o}}{RT}\;+\;\frac{\Delta S^{o}}{R}\;$$
(13)$$\Delta \;G^{o}=\Delta \;H^{o}-T\;\Delta \;S^{o}\;$$

The obtained ∆*H**^o^* values for the CFXN and MLG removal using CNPs indicated an exothermic sorption. Furthermore, ∆*G**^o^* implied the spontaneity of CFXN and MLG adsorption within the tested range of concentrations [58]. Additionally, the 10 and 25 mg L^−1^ solutions showed a significant increase in their ∆*G**^o^* as the temperature increased, indicating that the adsorption became highly temperature-sensitive at low pollutant concentrations. Furthermore, the negative ∆*S**^o^* findings revealed that the adsorptions of CFXN and MLG on the CNPs were favorable [59].

### 3.6. Investigation of Sorption Nature

In sorbent–sorbate systems, aromatic pollutants are often adsorbed through the π–π interaction, electron donation acceptance, and/or electrostatic attraction, but chemical bonding is also likely to occur [60]. The values of the FI factor (1/n < 1.0) revealed a reversible adsorption process, which implied physisorption. CFXN and MLG concentrations of 10 and 25 mg L^−1^ showed that ∆*H^o^* values were more than 80 kJ mol^−1^, contradicting the pure physisorption conclusion. Hence, FTIR was utilized to examine the functional groups of the adsorbed CFXN and MLG molecules [61]. Figure 8a demonstrates the vibrational bands resulting from the free CFXN compared with the adsorbed CFXN bands. The free CFXN showed a vibration band above 3000 cm^−1^, which could be attributed to the O-H stretching. The lump at 3300 cm^−1^ may belong to the N-H stretching vibration, which did not appear after the adsorption, implying a formation of an amide bond with the oxygen functional groups on the CNPs. Additionally, the adsorbed CFXN showed a carbonyl band at approximately 1640 cm^−1^, indicating an amide bond formation [36]. These results indicated that chemisorption had participated in removing CFXN. Nevertheless, at high pollutant concentrations, physisorption occurred alongside the former.

Figure 8b showed the FTIR results of the free MLG and adsorbed MLG molecules on the CNPs. The free MLG showed vibration bands between 700 and 850 cm^−1^ that could be assigned to out-of-plane C-H bending. The two peaks at approximately 1200 to 1400 cm^−1^ may belong to the geminal dimethyl group and/or C-N stretching vibration. The band at approximately 1590 cm^−1^ may be attributed to the C=C stretching vibration of the ring connected to the positive nitrogen in the MLG molecule. The peak at approximately 1690 cm^−1^ could be assigned to the C=N stretching vibration and/or C=C of the tert-substituted carbon. In addition, the C-H stretching vibrations of the methyl and aromatic rings appeared at approximately 2940 cm^−1^ and 3050 cm^−1^, respectively [36,41]. Because of the CNPs, the adsorbed MLG molecules exhibited vibration bands of lower intensities, but none of the peaks disappeared, indicating the absence of chemisorption. Hence, the possible sorbent–sorbate interactions may include an electrostatic attraction between the partially negative oxygen groups on the CNPs and the positive nitrogen on the MLG molecules. Additionally, hydrogen bonding occurred between the oxygen groups on the CNPs and MLG molecules, as well as a π–π interaction between their aromatic groups [62].

### 3.7. Application to Natural Water Samples

The ball-mill-fabricated CNPs were examined for removing CFXN and MLG from GWSs and SWSs. Figure 9a displays the removal efficiency of CNPs in treating contaminated GWSs and SWSs with 5.0 and 10.0 mg L^−1^ of each pollutant. The prepared CNPs removed CFXN and MLG entirely from the 5.0 mg L^−1^ contaminated GWS and SWS.

### 3.8. Regeneration and Reuse

The sorbent reusability is an economic factor, and it was essential for examining this task. Since the acidic pH below 4.0 affected the adsorption capability of CNPs, a 2.0 mol L^−1^ hydrochloric acid solution (2M HCl) was selected as a regeneration solution. The used CNPs were sonicated with 50 mL of 2M HCl for 15 min, filtered, and the process was repeated with 50 mL distilled water, then 20 mL ethanol. The regenerated CNPs were vacuum filtered, rinsed with 100 mL distilled water, activated at 150 °C for 60.0 min, and employed for the next round. Figure 9b illustrates the application of the used CNPs in removing CFXN and MLG. The CNPs showed an average removal percentage of 95.1% and 96.8% for CFXN and MLG during the four cycles and exhibited RSD values of 3.5% and 3.4%, respectively. Due to the chemisorption nature, a decrease in the removal percentage of the CNPs with CFXN was expected. Nevertheless, the CNPs possessed a consistently high performance during the reusability investigations.

## 4. Conclusions

SFSW was carbonization at 600 °C, ground with ball milling, and functionalized via an acid treatment. The prepared CNPs were characterized and tested for removing CFXN and MLG from aqueous solutions via batch experiments. The adsorption of both pollutants followed the PSFO kinetic model, and their adsorptions appeared to be controlled by the intraparticle diffusion step. CFXN and MLG adsorption on the CNPs fitted the LI and FI model, respectively. The sorptions of both pollutants on the CNPs were exothermic and spontaneous. FTIR was used to investigate functional groups’ appearance and/or disappearance from the adsorbed contaminants. In the case of MLG, the removal was mediated by physisorption, while for CFXN, the physicochemical process took place. The ball-mill-fabricated CNPs showed an excellent performance in removing CFXN and MLG from GWSs and SWSs, and possessed consistent efficiency during the recycle–reuse study. The complete removal of the 5 mg L^−1^ of both pollutants suggested this sorbent’s applicability to remediate water from these pollutants.

## Figures and Tables

**Figure 1 molecules-27-04577-f001:**
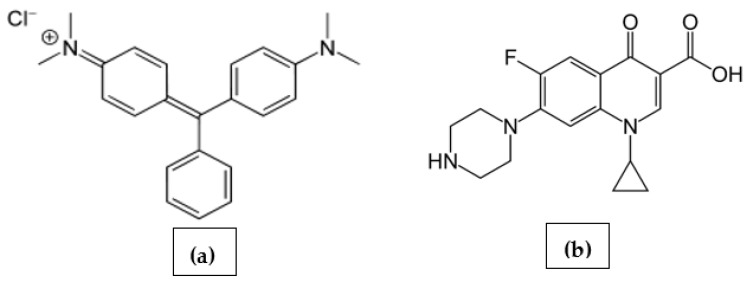
The chemical formulae of (**a**) malachite green dye and (**b**) ciprofloxacin antibiotic.

**Figure 2 molecules-27-04577-f002:**
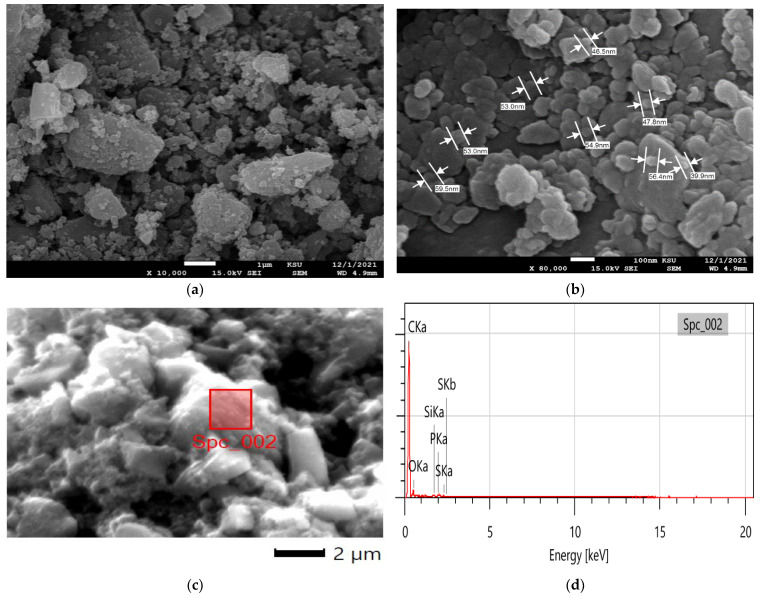
**(a**,**b**) SEM images; (**c**) electronic image corresponding to EDX analysis; (**d**) EDX spectrum; (**e**,**f**) TEM results of the ball-mill-fabricated CNPs.

**Figure 3 molecules-27-04577-f003:**
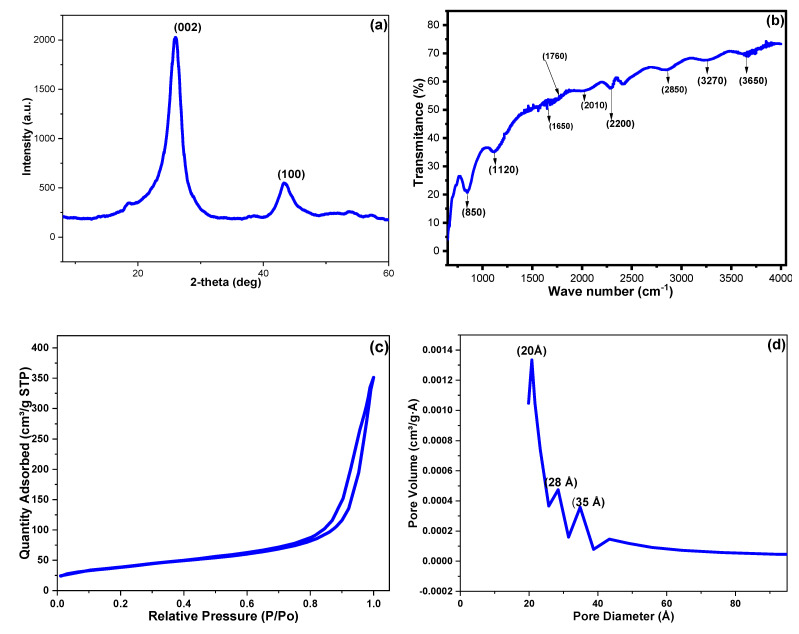
The obtained CNP results via (**a**) powder XRD diffraction spectroscopy, (**b**) FTIR spectroscopy, (**c**) nitrogen adsorption-desorption isotherm, and (**d**) pore diameter distribution.

**Figure 4 molecules-27-04577-f004:**
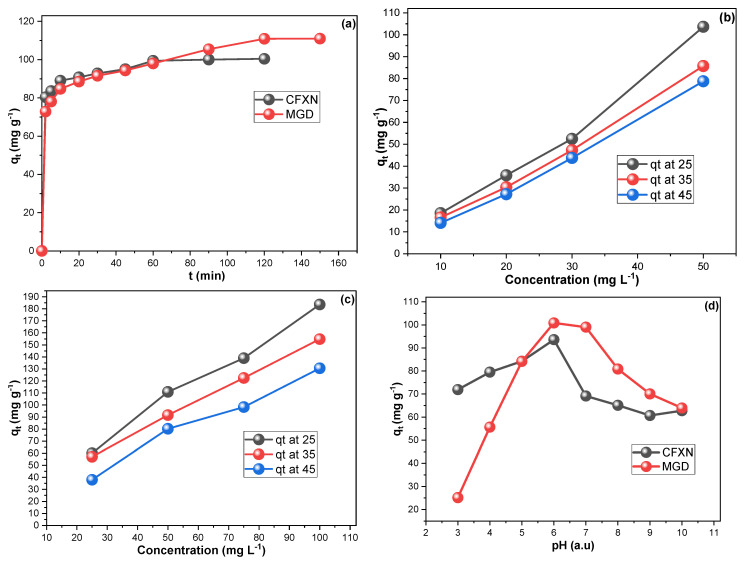
CFXN and MGD adsorption using CNPs, investigations of (**a**) the contact time study; (**b**,**c**) the impact of initial pollutant concentration on the adsorption of CFXN and MGD; and (**d**) the influence of the solution’s pH.

**Figure 5 molecules-27-04577-f005:**
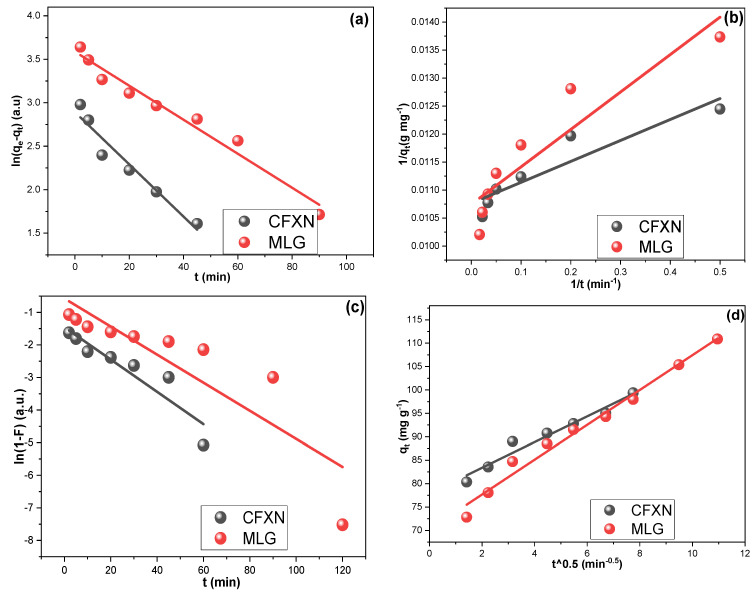
(**a**,**b**) present the adsorption rate order plots for PSFO and PSSO models for the adsorption of CFXN and MLG on the fabricated CNPs from 50.0 mg L^−1^ at 25 °C; (**c**,**d**) present the rate control mechanism investigations using LFDM and IPDM for CFXN and MLG adsorption on the fabricated CNPs from 50.0 mg L^−1^ at 25 °C.

**Figure 6 molecules-27-04577-f006:**
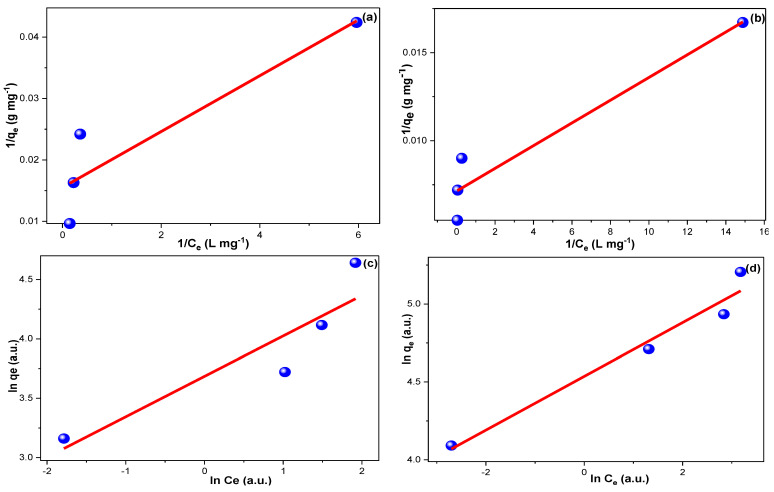
(**a**,**b**) present the Langmuir investigation for the adsorption of CFXN and MLG onto the CNPs, respectively; (**c**,**d**) show the Freundlich investigations for the same systems.

**Figure 7 molecules-27-04577-f007:**
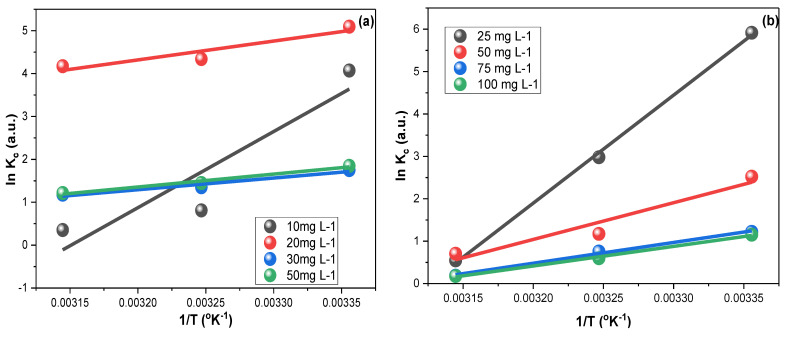
The thermodynamic results for the adsorption of (**a**) CFXN and (**b**) MLG on the CNPs within a temperature range of 25 °C to 45 °C using 10, 20, 30, and 50 mg L^−1^ CFXN solutions and 25, 50, 75, and 100 mg L^−1^ MLG solutions.

**Figure 8 molecules-27-04577-f008:**
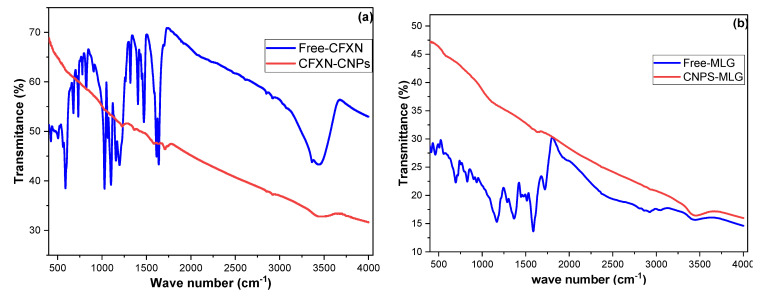
FTIR spectra of (**a**) free CFXN molecules and as adsorbed onto CNPs; (**b**) free MLG molecules and as adsorbed onto CNPs.

**Figure 9 molecules-27-04577-f009:**
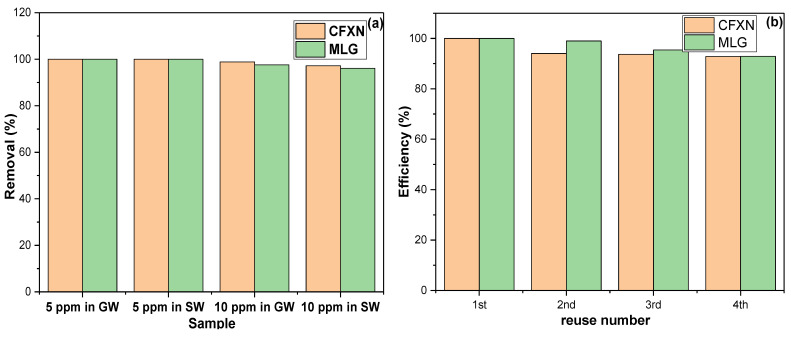
(**a**) The removal efficiency of CFXN and MLG from SW and GW samples with the prepared CNPs; (**b**) the reuse performance of the prepared CNPs in removing CFXN and MLG from aqueous solutions.

**Table 1 molecules-27-04577-t001:** The LI and FI parameters for the adsorption of CFXN and MLG on the ball-mill-fabricated CNPs at 25 °C using concentration ranges of 10 to 50 mg L^−1^ and 25 to 100 mg L^−1^ for CFXN and MLG, respectively.

Isotherm Model	LI			FI		
Pollutant	R^2^ (a.u)	K_L_ (L mg^−1^)	q_m_ (mg g^−1^)	R^2^ (a.u)	K_f_ (L mg^−1^)	1/n (a.u)
CFXN	0.846	64.419	3.414	0.828	39.836	0.340
MLG	0.923	140.071	11.068	0.961	93.296	0.173

**Table 2 molecules-27-04577-t002:** The thermodynamic results for the adsorption of CFXN and MLG on the CNPs within a temperature range of 25 °C to 45 °C using 10, 20, 30 and 50 mg L^−1^ CFXN solutions and 25, 50, 75, and 100 mg L^−1^ MLG solutions.

Fed Conc. (mg L^−1^)	ΔH^o^ (kJmol^−1^)	ΔS^o^ (kJmol^−1^)	ΔG^o^ (kJmol^−1^) 298 K	ΔG^o^ (kJmol^−1^) 308 K	ΔG^o^ (kJmol^−1^) 318 K	R^2^ (a.u)
CFXN						
10	−147.680	−0.465	−9.015	−4.362	0.291	0.854
20	−36.568	−0.081	−12.398	−11.587	−10.776	0.893
30	−22.658	−0.062	−4.252	−3.635	−3.017	0.958
50	−25.002	−0.069	−4.518	−3.831	−3.831	0.983
MLG						
25	−211.728	−0.662	−14.527	−7.910	−1.292	0.999
50	−72.003	−0.222	−5.921	−3.703	−1.486	0.937
75	−40.723	−0.126	−3.079	−1.816	−0.552	0.994
100	−38.464	−0.120	−2.812	−1.616	−0.420	0.996

## Data Availability

The data of this manuscript are available from the corresponding author upon reasonable request.

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
