# Peer review of "Optimization, Nature, and Mechanism Investigations for the Adsorption of Ciprofloxacin and Malachite Green onto Carbon Nanoparticles Derived from Low-Cost Precursor via a Green Route"

_molecules, 2022, doi:10.3390/molecules27144577_

Round 1

Reviewer 1 Report

In this work authors evaluated the sorption capacity of waste-derived carbon nanoparticles towards two selected organic contaminants, i.e. ciprofloxacin (CFXN) and malachite green (MLG). Carbon nanoparticles were prepared by pyrolysis of sunflower seed waste followed by ball-milling treatment. Most relevant textural features of the prepared carbon materials were characterized through conventional analytical technique (X-ray diffraction, Infrared spectroscopy, electron microscopy, N2 adsorption/desorption isotherms).

The paper could be of interest for researchers working in the wastewater remediation field and it well fits within several of the topical sections of Molecules. However, the quality of the manuscript needs to be improved before to be published on this journal.

Some comments are reported below.

1)      Title. The title is appropriate. Please, replace rout with route.

2)      Introduction. The introduction could be improved. Please, can authors discuss some examples of typical adsorbents reported in the literature for CFXN and MLG capture. I also suggest adding the molecular structure of these molecules and reporting some relevant physico-chemical properties (dielectric constant, boiling point).

3)      Experimental.

a) p.2 line 86-87 Please, add more details about surface area measurements, e.g. sample pre-treatment conditions, probe gas, model equations.

b) p.3 line 95 Please, specify the wavelength.

c) p.2 line 79 Please, spell out DW.

4)      Results and discussion

a) p.3, Figure 1. Authors are invited to comment EDX analysis results, in particular panels e and f of Fig.1 should be described and commented in the main text.

b) p.4 Please move equations 2-5 to the Experimental section.

c) Structure and surface characterization needs to be improved. Raman spectroscopy is a must for the characterization of graphitization degree of carbon materials. Similarly, the characterization of surface functional groups requires X-ray photoelectron spectroscopy.

d) The discussion on pH effect should be extended, by taking into account the isoelectric point of the sorbent and the pKa and pKb of the functional groups of the two adsorbate molecules.

e) p.12 line 286, please correct HCL in HCl.

Author Response

Reviewer 1

Comments and Suggestions for Authors

The paper could be of interest for researchers working in the wastewater remediation field and it well fits within several of the topical sections of Molecules. However, the quality of the manuscript needs to be improved before to be published on this journal.

Thank you for your valuable time reviewing this work, and we really appreciate your constructive comments.

1)      Title. The title is appropriate. Please, replace rout with route.

Done, additional adsorption references have been cited

2)      Introduction. The introduction could be improved. Please, can authors discuss some examples of typical adsorbents reported in the literature for CFXN and MLG capture. I also suggest adding the molecular structure of these molecules and reporting some relevant physico-chemical properties (dielectric constant, boiling point).

Done, the structures and some have been added to the introduction, and some physicochemical properties have been added in the discussion section.

3)      Experimental.

  1. a) p.2 line 86-87 Please, add more details about surface area measurements, e.g. sample pre-treatment conditions, probe gas, model equations.

Done.

  1. b) p.3 line 95 Please, specify the wavelength.

Done.

  1. c) p.2 line 79 Please, spell out DW.

Done.

4)      Results and discussion

  1. a) p.3, Figure 1. Authors are invited to comment EDX analysis results, in particular panels e and f of Fig.1 should be described and commented in the main text.

Done.

  1. b) p.4 Please move equations 2-5 to the Experimental section.

Done.

  1. c) Structure and surface characterization needs to be improved. Raman spectroscopy is a must for the characterization of graphitization degree of carbon materials. Similarly, the characterization of surface functional groups requires X-ray photoelectron spectroscopy.

Done, additional characterizations have been added.

  1. d) The discussion on pH effect should be extended, by taking into account the isoelectric point of the sorbent and the pKa and pKb of the functional groups of the two adsorbate molecules.

The pH effect has been elaborated.

  1. e) p.12 line 286, please correct HCL in HCl.

Done.

Reviewer 2 Report

1.       Revise the title of the paper there are some grammatical mistakes

2.       For nanomaterial size of the particles is of key importance which has not been mentioned in the abstract section.

3.       The techniques used to characterize the particles must be mentioned in abstract section.

4.       Line 23-24, “followed the Langmuirand the Fredulich isotherm models” is a senseless sentence

5.       Line 25-26,” Also, the adsorption nature was analyzed by FTIR, which revealed chemisorption for CFXN and physisorption for MLG” how FTIR could be helpful to determine the nature of adsorption process? Also, if first order kinetic is followed then how chemisorption is the process.

6.       In introduction the novelty statement is weak

7.       The literature cited is mostly irrelevant as suggestion some relevant work is: Novel Magnetite Nanocomposites (Fe3O4/C) for Efficient    Immobilization of Ciprofloxacin from Aqueous Solutions Through Adsorption Pretreatment and Membrane Processes. Water 2022, 14, 724. https://doi.org/10.3390/w14050724

Removal of ciprofloxacin from water through magnetic nanocomposite/membrane hybrid processes. Desalination and water treatment. 137: 260-272.

8.       Line 74, “50 g of the SFSW” a sentence never starts with numeral. Please write about or approximately before it.

9.       Line 75, “at 600 ℃ under” there should be no space between numerical value and temp unit as it is exception for other physical quantities space is required.

10.   Section 2.4, it should be separated into subheadings like isothermal study, kinetics etc.

11.   SEM images in figure is taken at 1 micrometer whereas the second one is at 100 nm the units should be the same. Also, from figure it seem the it is taken at high magnification as a 100 nm pictures are mostly blurred but here it clear, creating some doubts.

12.    The particle size estimated only small particles have been targeted, what about the big particles and lumps.

13.   SEM picture in figure C do not show any information about magnification etc.

14.   Figure 1 D, the edx graphs the peaks do not have labels.

15.   Figure 1 e, f I do not its significance to add them in the paper

16.   XRD analysis, “The diffraction peaks 110 at 2θâ—¦ of 26.02 and 43.42 can be assigned to (002) and (100).” These are peaks but the indices 002 and 100 represent which element or compound? Also, carbonaceous adsorbents are amorphous they do not show any peak in xrd, from where these peaks have been originated. If it was a composite then it was ok but in present case these peaks are misleading.

17.    Debye Scherer equation has been used but size has not mention in the respective section.

18.   Equilibrium time in adsorption kinetics is very important which have not been mentioned in the paper

19.   The R2 values in the graphs have not been mentioned, how one can decide the best fitting the isothermal and kinetics models.

20.   Table 2 how it is possible that the same time entropy, enthalpy and gibbs free energy values are negative. Also row first if the mentioned two parameters have negative value the third come out to be negative at 318k.

21.   Conclusion must be rephrased properly

22.   Reference style is also not uniform

Author Response

Comments and Suggestions for Authors

Thank you for your valuable time reviewing this work, and we really appreciate your constructive comments.

  1. Revise the title of the paper there are some grammatical mistakes

Done.

  1. For nanomaterial size of the particles is of key importance which has not been mentioned in the abstract section.

Done.

  1. The techniques used to characterize the particles must be mentioned in abstract section.

Done.

  1. Line 23-24, “followed the Langmuirand the Fredulich isotherm models” is a senseless sentence

Done, the sentence has been rephrased.

  1. Line 25-26,” Also, the adsorption nature was analyzed by FTIR, which revealed chemisorption for CFXN and physisorption for MLG” how FTIR could be helpful to determine the nature of adsorption process? Also, if first order kinetic is followed then how chemisorption is the process.

Done, the sentence has been rephrased.

  1. In introduction the novelty statement is weak.

Done, the statement has been improved.

  1. The literature cited is mostly irrelevant as suggestion some relevant work is: Novel Magnetite Nanocomposites (Fe3O4/C) for Efficient    Immobilization of Ciprofloxacin from Aqueous Solutions Through Adsorption Pretreatment and Membrane Processes. Water2022, 14, 724. https://doi.org/10.3390/w14050724

Removal of ciprofloxacin from water through magnetic nanocomposite/membrane hybrid processes. Desalination and water treatment. 137: 260-272.

Done.

  1. Line 74, “50 g of the SFSW” a sentence never starts with numeral. Please write about or approximately before it.

Done.

  1. Line 75, “at 600 â„ƒunder” there should be no space between numerical value and temp unit as it is exception for other physical quantities space is required.

Done.

  1. Section 2.4, it should be separated into subheadings like isothermal study, kinetics etc.

Done.

  1. SEM images in figure is taken at 1 micrometer whereas the second one is at 100 nm the units should be the same. Also, from figure it seem the it is taken at high magnification as a 100 nm pictures are mostly blurred but here it clear, creating some doubts.

Of course, you understand that the scale is auto-adjusted according to the degree of magnification. The first SEM image is an overview of the prepared particles, and for that, it was taken at low magnification, which makes the scale micro. The second one is a high magnification image of the same powder.

  1. The particle size estimated only small particles have been targeted, what about the big particles and lumps.

The size of the large particles has been added.

  1. SEM picture in figure C do not show any information about magnification etc.

Done, another image has been placed.

  1. Figure 1 D, the edx graphs the peaks do not have labels.

Done, the EDX analysis was conducted again with, and the obtained graph was added.

  1. Figure 1 e, f I do not its significance to add them in the paper.

Done, the figures have been removed.

  1. XRD analysis, “The diffraction peaks 110 at 2θâ—¦ of 26.02 and 43.42 can be assigned to (002) and (100).” These are peaks but the indices 002 and 100 represent which element or compound? Also, carbonaceous adsorbents are amorphous they do not show any peak in xrd, from where these peaks have been originated. If it was a composite then it was ok but in present case these peaks are misleading.

Done, the sentence has been rephrased.

  1. Debye Scherer equation has been used but size has not mention in the respective section.

Done.

  1. Equilibrium time in adsorption kinetics is very important which have not been mentioned in the paper.

It was mentioned; please find the second sentence in section 3.2.

  1. The R2values in the graphs have not been mentioned, how one can decide the best fitting the isothermal and kinetics models.

All regression data was collected in the tables as mentioned in the main text.

  1. Table 2 how it is possible that the same time entropy, enthalpy and gibbs free energy values are negative. Also row first if the mentioned two parameters have negative value the third come out to be negative at 318k.

The was no mistake in the calculations; the entropy was a negative fraction value which yields a little positive value when multiplied by the temperature values leading to a third negative value.

  1. Conclusion must be rephrased properly

Done.

  1. Reference style is also not uniform

The references style was managed via the Endnote program.

Round 2

Reviewer 1 Report

Accept as it is

Reviewer 2 Report

ok